# Congenital Hyperinsulinism Caused by Mutations in *ABCC8* Gene Associated with Early-Onset Neonatal Hypoglycemia: Genetic Heterogeneity Correlated with Phenotypic Variability

**DOI:** 10.3390/ijms25105533

**Published:** 2024-05-19

**Authors:** Lăcrămioara Ionela Butnariu, Delia Andreia Bizim, Gabriela Păduraru, Luminița Păduraru, Ștefana Maria Moisă, Setalia Popa, Nicoleta Gimiga, Gabriela Ghiga, Minerva Codruța Bădescu, Ancuta Lupu, Ioana Vasiliu, Laura Mihaela Trandafir

**Affiliations:** 1Department of Medical Genetics, Faculty of Medicine, “Grigore T. Popa” University of Medicine and Pharmacy, 700115 Iasi, Romania; setalia_popa@yahoo.com; 2Departament of Diabetes, Saint Mary’s Emergency Children Hospital, 700309 Iasi, Romania; 3Department of Mother and Child, Faculty of Medicine, “Grigore T. Popa” University of Medicine and Pharmacy, 700115 Iasi, Romania; paduraru.gabriela@umfiasi.ro (G.P.); stefana-maria.moisa@umfiasi.ro (Ș.M.M.); nicoleta.chiticariu@umfiasi.ro (N.G.); gabriela.ghiga@umfiasi.ro (G.G.); anca_ign@yahoo.com (A.L.); laura.trandafir@umfiasi.ro (L.M.T.); 4Department of Mother and Child, Division Neonatology, Faculty of Medicine, “Grigore T. Popa” University of Medicine and Pharmacy, 700115 Iasi, Romania; luminita.paduraru@umfiasi.ro; 5Department of Internal Medicine, “Grigore T. Popa” University of Medicine and Pharmacy, 16 University Street, 700115 Iasi, Romania; minerva.badescu@umfiasi.ro; 6Department of Morphofunctional Sciences II, Grigore T. Popa University of Medicine and Pharmacy, 700115 Iasi, Romania; ioana.vasiliu@umfiasi.ro

**Keywords:** hyperinsulinemic hypoglycemia, *ABCC8* gene, mutation, ATP-sensitive potassium channel, genetic heterogeneity, phenotypic variability

## Abstract

Congenital hyperinsulinism (CHI) is a rare disorder of glucose metabolism and is the most common cause of severe and persistent hypoglycemia (hyperinsulinemic hypoglycemia, HH) in the neonatal period and childhood. Most cases are caused by mutations in the *ABCC8* and *KCNJ11* genes that encode the ATP-sensitive potassium channel (K_ATP_). We present the correlation between genetic heterogeneity and the variable phenotype in patients with early-onset HH caused by *ABCC8* gene mutations. In the first patient, who presented persistent severe hypoglycemia since the first day of life, molecular genetic testing revealed the presence of a homozygous mutation in the *ABCC8* gene [deletion in the *ABCC8* gene c.(2390+1_2391-1)_(3329+1_3330-1)del] that correlated with a diffuse form of hyperinsulinism (the parents being healthy heterozygous carriers). In the second patient, the onset was on the third day of life with severe hypoglycemia, and genetic testing identified a heterozygous mutation in the *ABCC8* gene c.1792C>T (p.Arg598*) inherited on the paternal line, which led to the diagnosis of the focal form of hyperinsulinism. To locate the focal lesions, (18)F-DOPA (3,4-dihydroxy-6-[^18^F]fluoro-L-phenylalanine) positron emission tomography/computed tomography (PET/CT) was recommended (an investigation that cannot be carried out in the country), but the parents refused to carry out the investigation abroad. In this case, early surgical treatment could have been curative. In addition, the second child also presented secondary adrenal insufficiency requiring replacement therapy. At the same time, she developed early recurrent seizures that required antiepileptic treatment. We emphasize the importance of molecular genetic testing for diagnosis, management and genetic counseling in patients with HH.

## 1. Introduction

Congenital hyperinsulinism (CHI) has an estimated incidence of 1/50,000 live births and is the most common cause of severe and persistent hypoglycemia (hyperinsulinemic hypoglycemia, HH) in the neonatal, infancy and childhood periods [1,2,3]. Early diagnosis and treatment positively influence the prognosis, preventing permanent brain damage. Most commonly, CHI is the consequence of loss-of-function (LOF) mutations in the *ABCC8* (SUR1) and *KCNJ11* (Kir6.2) genes located on chromosome 11p15.1, which encode the adenosine triphosphate (ATP)-sensitive potassium channel (K_ATP_) of pancreatic β cells [4]. Mutations of other genes involved in the regulation of insulin secretion are rarely identified: *GCK, GLUD1, HADH, SLC16A1, HNF4A, HNF1A, UCP2, HK1* and *PGM1* (Figure 1) [5].

K_ATP_-CHI is associated with hyperplasia of Langherhans islets, which can be diffuse (all pancreatic β cells are affected) or focal (localized islet dysfunction (30–40% of all CHI cases)), with a correlation between the histological type (phenotype) and the genetic defect present (genotype) [5].

The two histopathological forms cannot be distinguished clinically. The best way to differentiate is by performing (18)F-dihydroxyphenylalanine (18F-DOPA) positron emission tomography/computed tomography (PET/CT) [6]. Focal CHI begins at older ages and frequently associates with hypoglycemic seizures compared to the diffuse form [5].

Patients with diffuse CHI frequently present homozygous recessive or a compound heterozygote mutation in the *ABCC8* or *KCNJ11* genes (which encode the SUR/Kir6.2 components of the K_ATP_ channel in pancreatic β cells) [5,7].

The molecular mechanism in focal CHI follows the “two-hit” model described by Knudson [8]. It involves the presence of a heterozygous, paternally inherited mutation in the *ABCC8* or *KCNJ11* genes and the appearance in some pancreatic cells of the second mutation in the chromosomal 11p15 region of maternal origin with loss of heterozygosity (LOH). Another possible mechanism would be paternal uniparental isodisomy of chromosome 11p15.5 and the absence of the same region of maternal origin in focal lesions [5,9]. Mutations with loss of heterozygosity (LOH) in pancreatic somatic cells will determine the unbalanced expression of the imprinted genes (paternal *IGF2*, maternal *H19* and *CDKN1C*) from the chromosomal 11p15.5 region, which regulates cell growth, with the appearance of focal adenomatous hyperplasia [10,11]. The presence of the paternally inherited, heterozygous K_ATP_ mutations has a predictive value for focal CHI in 94% of cases [10,12]. In the focal form of CHI, the lesions are unique. To date, few cases with multifocal lesions have been reported. The differential diagnosis between the two forms of CHI (focal and diffuse) is important for therapeutic approach. First-line drug treatment consists of oral diazoxide, glucagon, somatostatin analogues. In focal CHI, the curative treatment involves the surgical excision of the focal lesion, while in diffuse CHI, which does not respond to drug treatment, the symptoms can only be improved through subtotal pancreatectomy with the risk of complications, such as exocrine pancreatic insufficiency and diabetes mellitus [13,14,15].

We present the genotype–phenotype correlation in patients with early-onset hyperinsulinemic hypoglycemia (HH) caused by different mutations in the *ABCC8* gene. The first patient presents a homozygous autosomal recessive mutation in the *ABCC8* gene associated with a diffuse CHI (both parents being healthy carriers of the same mutation); in the second patient, a paternally inherited, heterozygous *ABCC8* mutation was identified that led to a focal CHI.

## 2. Results

Genotype–Phenotype Correlation in Patients with Early-Onset Hyperinsulinemic Hypoglycemia (HH) Caused by Different Mutations in the *ABCC8* Gene.

### 2.1. Patient 1

We present the patient A.I.S., currently 3 months old, who was transferred from the neonatology department to the pediatric clinic at the age of 10 days due to persistent episodes of severe hypoglycemia. The baby comes from a noncomplicated pregnancy of a young and non-consanguineous couple. The mother was 28 years old, gesta II, para II, with no risk factors for diabetes, neither before nor during gestation, and no drug consumption or other treatments until delivery. She has one previous child, healthy and 4 years old, born at term with 2950 g and an Apgar score of 9. The father was 31 years old, healthy, with no history of chronic diseases.

The birth occurred at term (gestational age of 38 weeks) with a birth weight of 3140 g, pelvic presentation, acute fetal distress, an Apgar score of 6 at 1 min and 8 at 5 min and prolapse of the left lower limb, necessitating hospitalization of the child in the neonatal intensive care unit. Transfontanelle ultrasonography identified a bilateral subependymal hemorrhage. From the first day of life, the baby presented severe hypoglycemia with blood glucose values of 13 mg/dL at 11 h postnatal and 30 mg/dL at 26 h of life, requiring PEV with 12.5% glucose from the first days of life.

At blood glucose values below 50 mg/dL, plasma insulin showed a value of 41.35 uIU/mL (normal value: 3–25 uIU/mL) and C-peptide of 4.92 ng/mL (normal value: 0.2–4.4 ng/mL)—both increased; serum cortisol and thyroid hormones were within normal limits with an increased growth hormone of 133.67 uIU/mL (normal value: 0–55 uIU/mL) (Table 1).

A form of congenital hyperinsulinism was suspected, and a genetic consultation was requested, which recommended molecular genetic testing (gene panel).

In the pediatric unit, the treatment with diazoxide was initiated in a progressively increasing dose with frequent monitoring of blood glucose levels on a blood glucose meter, and at a dose of 10 mg/kg/day, hydrochlorothiazide was also added.

Glucose IV infusions were not required, and the child was fed orally every 2.5–3 h. At the maximum dose of diazoxide, hypoglycemic episodes became rarer, but severe hypoglycemia persisted, 1–2 episodes/day with a flat glycemic curve on a continuous glycemic monitoring system; therefore, we switched to the second-line medication, a rapid somatostatin analogue (sandostatin). Depending on the glycemic profile, the dose of the rapid somatostatin analogues was increased up to 25 mcg/kg/day in three subcutaneous doses under which remission of severe hypoglycemia was obtained with feeding at 2.5–3 h intervals without immediate adverse effects and with mild hypoglycemia (4%) on continuous glycemic monitoring (CGM)/7 days.

We tried to space out the meals, but severe hypoglycemia still occurred. In evolution, the initial total dose of the rapid somatostatin analogues did not require adjustment with the dose related to the child’s weight decreasing over time to 16.8 mg/kg/day at the age of 3 months.

Abdominal magnetic resonance imaging (MRI) did not identify pathological changes in the pancreas.

Molecular genetic testing (Hypoglycemia, Hyperinsulinism and Ketone Metabolism Panel, Blueprint Genetics Laboratory) identified a homozygous deletion in the *ABCC8* gene, c.(2390+1_2391-1)_(3329+1_3330-1)del, which encompasses exons 20–26; the result correlating with diffuse CHI. Genetic testing of the parents revealed that both parents are heterozygous for the same mutation identified in the child (Figure 2).

Since, under the current therapy, the infant did not present apparent hypoglycemic episodes, the surgical team decided to postpone the surgery, taking into account the young age of the child and the intra- and postoperative possible risks; in this case, a subtotal pancreatectomy will probably be recommended.

### 2.2. Patient 2

The second patient is also a girl, D.M.S., aged 1 year and 3 months, who was evaluated in the pediatric clinic at the age of 14 days for persistent severe hypoglycemia. The child comes from the first pregnancy of a young and non-consanguineous couple (Figure 3).

The pregnancy progressed apparently normally, being monitored by ultrasound. There were no risk factors for maternal diabetes before or during pregnancy, and no drugs were used during pregnancy. The birth occurred at term (gestational age of 38 weeks) by caesarean section, cranial presentation; birth weight was 2700 g, the Apgar score was 6 at 1 min with good adaptation to extrauterine life but with glycemic values of 60 mg/dL in the first days of life, and the child was discharged at 48 h of life.

At 3 days old, the parents initially contacted the territorial neonatology department because the child refused to eat, on which occasion a severe hypoglycemia was detected. A neonatal sepsis was initially suspected, but the persistence of hypoglycemia up to a value of 17 mg/dL raised the suspicion of a possible congenital hyperinsulinism; the child was later admitted to the pediatric clinic.

Biochemical investigations revealed a hypoglycemia of 41 mg/dL, respectively 46 mg/dL, a plasma insulin level of 15.28 uIU/mL, respectively 15.48 uIU/mL (normal value: 3–25 uIU/mL), and C-peptide of 2.2 ng/mL (normal value: 0.9–7.1 ng/mL), respectively 2.01 ng/mL, in two consecutive days (Table 1).

Thyroid hormones and growth hormone were simultaneously dosed with values within normal limits; on hypoglycemia, the plasma cortisol level was low (2.27 mcg/mL) (normal value: 4.3–22.4 mcg/mL). Plasma ACTH dosage was recommended; low values below 5 pg/mL were detected (normal value: 5–46 pg/mL), which suggested a central adrenal insufficiency.

A genetic consultation was requested, which recommended molecular genetic testing (gene panel for hypoglycemia), later performed at the Invitae laboratory, which identified a heterozygous *ABCC8* c.1792C>T (p.Arg598*) mutation present in the child and his father. This result raises the suspicion of a focal form of congenital hyperinsulinism (Figure 3).

The treatment with diazoxide and the attack dose of hydrocortisone was initiated, later in the substitution dose. The dose of diazoxide was progressively increased, associating hydrochlorothiazide at doses over 10 mg/kg/day of diazoxide, but severe hypoglycemia was maintained. A rapid somatostatin analogue (sandostatin) was then added up to a dose of 25 mcg/kg/body weight in four subcutaneous doses with a decrease in severe episodes of hypoglycemia under this treatment; after a month and a half of treatment with rapid somatostatin analogues, hypoglycemia was no longer evident during intermittent blood glucose monitoring, even allowing for a 6 h fasting period.

At the ages of 10 days and 1 month, the child presented a tonic-clonic seizure in the left hemibody of short duration, repeated, and antiepileptic therapy was instituted.

An urgent craniocerebral computed tomography was performed, which identified the presence of symmetrical hypodense areas occipital and posterior parietal bilaterally ischemic sequelae and, at the level of the pituitary gland, an inhomogeneous and hypocapturing nodule of 3/3 mm in the antero-inferior portion. Subsequently, a brain MRI performed at 7 months of age revealed a normal appearance.

The result of the genetic test raised the suspicion of focal CHI with a recommendation to perform a (18) F-DOPA PET/CT (3,4-dihydroxy-6-[^18^F]fluoro-L-phenylalanine) positron emission tomography scan/computed tomography.

Taking into account that, in Romania, there was no possibility of carrying out this investigation, they were directed to a specialized clinic abroad, but the parents have so far refused this investigation.

In the evolution, it was not necessary to increase the initial total dose of the rapid somatostatin analogues because, during intermittent blood glucose monitoring, hypoglycemia was not identified (declaratively).

At the age of 9 months, the endocrinological evaluation in a specialized clinic revealed the normalization of plasma levels of ACTH and cortisol and Hb A1c = 4.9%. Glycemic monitoring during the respective hospitalization did not reveal hypoglycemia at the dose of rapid somatostatin analogues initiated in infancy, at present 11 mcg/kg body weight.

At the clinical examination, the child showed inappropriate weight gain and moderate neuromotor retardation as well as the recurrence of epileptic seizures, necessitating the adjustment of the antiepileptic treatment.

## 3. Discussion

In both children, the onset of hypoglycemia was early, from the first days of birth, being severe. First-line treatment with diazoxide was initiated in the first two weeks of life, but both children were unresponsive, switching to a second-line medication (somatostatin).

To control hypoglycemia, a higher dose of somatostatin than recommended by international guidelines was initially required; the dose which, however, did not have to be adjusted to the child’s weight, maintaining the initial dose.

The second child also had a central adrenal insufficiency that required substitution treatment. At the same time, he developed early recurrent seizures that required the initiation of anticonvulsant therapy.

We mention that, in Romania, it is not possible to perform (18)F-DOPA PET/CT, which is necessary in this case for the localization of focal lesions. This form of hyperinsulinism would have required surgical treatment, which could have been curative if it had been instituted early.

Both children presented an inappropriate weight curve with an early weight deficit in case 1 with a weight index of 0.72 at 3.5 months despite an appropriate food intake. The anthropometric data of the second patient were within normal limits initially; later, a progressive stature–weight growth delay became constant with stature value of −2.16 standard deviation (SD) and weight value of −2.16 SD at the age of 1 year. The second child also had a moderate psychomotor retardation.

The two cases presented demonstrated genetic heterogeneity (different mutations in the *ABCC8* gene) associated with phenotypic variability. Thus, in the first case, a homozygous mutation of the *ABCC8* gene was correlated with diffuse CHI, while in the second case, the patient had a paternally inherited heterozygous *ABCC8* mutation, which was correlated with a possible focal CHI. The presented data are consistent with those from the specialized literature.

In the first patient, del/dup copy number variation (CNV) analysis using the Blueprint Genetics (BpG) Hypoglycemia, Hyperinsulinism and Ketone Metabolism Panel identified a homozygous deletion, *ABCC8* c.(2390+1_2391-1)_(3329+1_3330-1)del, encompassing exons 20–26 of *ABCC8*. This deletion is estimated to cover the genomic region 11:17428108-17435103 and is approximately 6995 base pairs in size. Gross deletions encompassing the protein coding regions of the *ABCC8* gene have not been reported in the Genome Aggregation Database control cohorts (gnomAD SVs v2.1) [16]. In the Human Gene Mutation Database (HGMD) professional database (2023.3), there are currently more than 20 gross *ABCC8* deletions (mostly partial gene deletions) reported in association with *ABCC8*-related diseases [17]. For instance, De Franco et al. identified several large *ABCC8* deletions in patients with hyperinsulinism, including a deletion of exons 22–26 that was reported to be compound heterozygous (the second variant was not specified) (PMID: 32027066) [18].

In the second patient, the genetic analysis included sequencing and del/dup analysis in the case of 120 genes associated with hypoglycemia (Invitae Hypoglycemia panel). Thus, it was proven that the patient had a heterozygous pathogenic variant in the *ABCC8* c.1792C>T (p.Arg598*) of paternal origin, suggestive of focal CHI.

The Arg598Ter variant has been reported in over 10 individuals with congenital hyperinsulinism [19,20]. Approximately 0.006% of African Americans are healthy heterozygous carriers for this single nucleotide polymorphism (SNP) with National Center for Biotechnology Information (NCBI) dbSNP ID rs13932856 [21], according to The Genome Aggregation Database (gnomAD; http://gnomad.broadinstitute.org, accessed on 7 March 2024) [16].

This variant (Arg598Ter) was also reported as pathogenic in ClinVar (Variation ID: 434056) (https://www.ncbi.nlm.nih.gov/clinvar/variation/434056/, accessed on 7 March 2024) [22]. Of the eleven affected individuals, at least four were compound heterozygotes carrying a reported pathogenic variant in trans, increasing the likelihood that the Arg598Ter variant is pathogenic [23,24,25,26,27].

In vitro functional studies provided evidence that the Arg598Ter variant may slightly affect protein function [26,28]. This variant causes a premature stop codon at position 598, leading to a truncated or absent protein. Paternally inherited loss-of-function (LOF) *ABCC8* mutations represent a known mechanism in autosomal-recessive hyperinsulinemic hypoglycemia [29,30].

Carriers of a single heterozygous pathogenic familial hyperinsulinemic hypoglycemia, type 1 (FHI)-associated variant inherited from the father may be at risk for focal FHI. Focal FHI occurs when a single pathogenic FHI-associated variant is inherited from a carrier father and a second change occurs in only some of the pancreatic cells, causing the loss of the normal maternal gene. In the area of the pancreas in which only the paternal FHI gene is represented, insulin is overproduced and may cause hyperinsulinism of variable severity [29,30].

In the second case, the patient’s father presented the same *ABCC8* (Arg598Ter) heterozygous mutation as the child but did not present symptoms of hypoglycemia. The allelic expression imbalance (AEI) could explain the variable phenotypic expressivity in this case, the father and the child presenting different phenotypes, although the same *ABCC8* mutation was present. AEI refers to the different gene expression in intensity of the two alleles of the genes that encode the same protein. Initially, it was thought that the expression of maternal and paternal alleles is balanced, and this balance could reduce the effect of recessive mutations. However, several mechanisms are involved in the regulation of gene expression, including epigenetic ones. Subsequent research demonstrated that AEI occurred when the expression of one of the alleles was inhibited or exacerbated in the case of post-transcriptional degradation of mature mRNA [30]. The existence of AEI in the case of the *ABCC8* gene will be elucidated through future studies.

*ABCC8* (OMIM 600509) encodes ATP-binding cassette transporter subfamily C member 8 member 8, which is expressed in pancreatic β cells and in the nervous system [4]. Together with the proteins encoded by the *KCNJ11, KCNJ8* and *ABCC9* genes, *ABCC8* forms the ATP-sensitive potassium channel (K_ATP_) that detects metabolic changes in pancreatic β cells and regulates insulin secretion. LOF mutations in the *ABCC8* or *KCNJ11* genes lead to K_ATP_ channel dysfunction and hyperinsulinism. Depolarization of the cell membrane occurs even in the absence of an increased intracellular ATP/ADP ratio, initiating the insulin secretion cascade, even in the absence of glucose [31].

In the case of the *ABCC8* gene, a genetic heterogeneity correlated with the phenotypic variability is described with over 890 variants in *ABCC8* annotated as disease-causing mutations (DCM) in the HGMD Professional variant database, including both missense and truncating variants (nonsense, frameshift, variants affecting splicing, gross deletions). Of these, over 400 are associated with HH, and at least 14 mutations have been associated with permanent neonatal diabetes mellitus (PNDM) [19,32].

Pathogenic mutations in the *ABCC8* gene cause autosomal dominant and recessive FHI (OMIM 256450) and dominant leucine-sensitive hypoglycemia of infancy (OMIM 240800) [4]. Other pathogenic *ABCC8* variants are associated with autosomal dominant type 2 diabetes mellitus (OMIM 125853), PNDM (OMIM 606176) and transient neonatal diabetes mellitus (TNDM) (OMIM 610374) [4].

Although pathogenic variants in *ABCC8* are more frequently associated with permanent and transient neonatal diabetes, late-onset cases are described. It is proven that autosomal dominant hyperinsulinism caused by LOF *ABCC8* mutations develops reduced glucose tolerance and, in some cases, diabetes mellitus [12,32,33].

Gain-of-function (GOF) missense mutations in the *ABCC8* gene are detected in cases of PNDM. K_ATP_ channels carrying these mutations lose regulatory inhibition by ATP [34].

Thus, in PNDM patients, persistent hyperglycemia is caused by loss of pancreatic β-cell membrane excitability to glucose and loss of pancreatic insulin. Glucose normally increases β-cell excitability by inhibiting K_ATP_ channels, opening voltage-dependent calcium channels, increasing intracellular calcium [Ca^2+^]_i_, which triggers insulin secretion [35,36].

*ABCC8* mutations are detected in more than 45% of FHI cases [37,38]. Recessive LOF *ABCC8* mutations are detected in patients with FHI/congenital hyperinsulinism, in which heterozygous individuals are healthy carriers [39].

The phenotypic severity in FHI cases varies from severe hypoglycemia with neonatal onset, difficult to treat, to milder manifestations of the disease with reduced symptoms, which begin in childhood; in their case, there are difficulties related to the diagnosis of hypoglycemia [40].

K_ATP_-channel inactivating mutations in *ABCC8* associated with mutations in the *KCNJ11* gene cause 97% of cases of diazoxide-unresponsive hyperinsulinism [40,41]. An autosomal recessive and, more rarely, an autosomal dominant transmission are detected most frequently, but de novo mutations have also been reported [40].

In approximately 97% of FHI cases detected in the Ashkenazi Jewish population, two *ABCC8* founder mutations (c.3989-9g>a and p.F1387del) were detected [42]. The pathogenic variant c.3989-9G>A has been identified in several different ethnic groups, suggesting that this is a hotspot mutation. Homozygous recessive KA_TP_-channel mutations associated with impaired insulin secretion are associated with diffuse pancreatic islet damage. These allelic variants can also cause focal adenomatosis of β cells when a paternally derived K_ATP_ variant becomes expressed through embryonic loss of heterozygosity for the maternal allele in a clone of β cells [42].

More than 40% of cases with FHI have pancreatic adenomatous hyperplasia involving a limited region of the pancreas (focal CHI). In this case, the transmission is autosomal dominant but only manifests when the pathogenic variant occurs on the paternally derived allele and a somatic event results in the loss of the maternal allele in a β-cell precursor [42]. The clinical manifestations are similar to those of autosomal-recessive FHI-K_ATP_, but the genetic and therapeutic aspects are clearly different. In the form of autosomal-dominant FHI, the onset is after the age of 6–9 months, and the clinical manifestations are less severe than in the recessive form and usually respond to treatment with diazoxide [39].

The recessive forms of diffuse or focal FHI are associated postnatally with severe hypoglycemia that does not respond to treatment with diazoxide or octreotide, requiring surgical treatment (subtotal pancreatectomy) [43].

Sporadic forms of HH are associated with moderate/severe episodes of hypoglycemia and hyperinsulinism evident from the first days of life and usually have a poor response to treatment, but the prognosis improves after partial pancreatectomy [19].

The drug treatment of choice in CHI is diazoxide, which binds to the intact SUR1 component of K_ATP_ channels. It keeps K_ATP_ channels open, preventing pancreatic β-cell membrane depolarization and insulin secretion. There are studies that have proven that diazoxide is usually ineffective in treating patients with CHI caused by recessive biallelic mutations in the genes encoding the K_ATP_ channel (*ABCC8/KCNJ11*) associated with the diffuse form of the disease [44,45] as well as in the case of the presence of the monoallelic focal form of CHI [45,46].

Homozygous recessive mutations in *ABCC8* and *KCNJ11* are usually null mutations or are amino acid substitutions that prevent channel trafficking to the plasma membrane, causing persistent plasma membrane depolarization and insulin release. Diazoxide acts as an agonist of the K_ATP_ channel, suppressing insulin secretion, thus preventing membrane depolarization; therefore, most patients who have this type of K_ATP_ gene mutation do not respond to diazoxide treatment [45,46]. Some in vitro studies have shown that mutations causing diazoxide-unresponsive hyperinsulinism produce SUR1 subunits that can form channels with Kir6.2 that normally traffic to the plasma membrane, as do dominant mutations associated with a favorable response to diazoxide. However, mutations associated with a lack of response to diazoxide are associated with a more severe impairment of the responses of expression channels to activation by diazoxide and MgADP [47].

The results of our study are consistent with the data from the literature because both the first patient (who presented a homozygous variant of the *ABCC8* gene) and the second patient (in whom a pathogenic heterozygous variant of the *ABCC8* gene transmitted on the paternal line was identified) showed resistance to the first-line therapy with diazoxide, and it was necessary to replace it with somatostatin. In a large study, Kapoor et al. analyzed the genotype–phenotype correlations in the case of 300 patients with CHI [45]. Thus, the most common genetic causes of CHI were *ABCC8/KCNJ11* mutations (*n* = 109, 36.3%). In 87.6% (92 patients) of the 105 patients who did not respond to diazoxide, an *ABCC8/KCNJ11* mutation was identified; a total of 63 patients had a homozygous recessive genotype, and 4 patients were heterozygous carriers of dominantly inherited mutations. A paternally transmitted variant of the *ABCC8/KCNJ11* genes was identified in 23 of the diazoxide-unresponsive patients, six of them presenting a diffuse CHI [45]. The results obtained in the mentioned study suggested that, from a clinical point of view, the vast majority of patients who do not respond to diazoxide therapy are likely to have mutations in the *ABCC8/KCNJ11* genes. The phenotypes of heterozygous patients were different from that of homozygotes for *ABCC8* mutations. Heterozygous carriers of a mutation in the *ABCC8* gene were either responsive or unresponsive to diazoxide therapy, in the latter case, having a dominant form of CHI [45].

## 4. Genetic Counseling

In the case of the first patient, genetic testing of the parents was indicated, which revealed that both are healthy carriers of the mutation present in the child [*ABCC8* c.(2390+1_2391-1) (3329+1_3330-1)del]. Their risk of having a new affected pregnancy is 25%, taking into account the autosomal-recessive transmission of the disease. In the second case, the risk of the couple, in which the father is heterozygous for the *ABCC8* c.1792C>T (p.Arg598*) mutation, having a new pregnancy that inherits the paternal mutation is 50%, which correlates with the autosomal-dominant pattern of inheritance. The probability that a sibling of a child with focal CHI will inherit the paternal *ABCC8* mutation is 50%, but the probability that he will also have somatic paternal uniparental disomy (UPD) for chromosome 11p15.5 is low [25].

In addition, heterozygous carriers of a mutation in the *ABCC8* gene require long-term monitoring, as they present an increased risk for diabetes mellitus [43,48].

## 5. Materials and Methods

We studied the correlations between different mutations in the *ABCC8* gene (genetic heterogeneity) and the variable phenotype in the case of two patients with HH diagnosed with different forms of the disease (focal form, respectively diffuse form of CHI). Molecular genetic testing (gene panel for hypoglycemia) of the patients and their parents was performed at laboratories abroad (Blueprint Genetics and Invitae); two different *ABCC8* variants were identified, correlated with the histological type of the disease.

The patients’ genomic DNA was extracted from peripheral blood using the QIAamp DNA Blood Mini Kit (Qiagen, Hilden, Germany). Targeted next-generation sequencing (NGS) and data analysis were performed using Illumina technology, in both cases being analyzed panels of genes associated with CHI (Blueprint Genetics (BpG) Hypoglycemia, Hyperinsulinism and Ketone Metabolism Panel, respectively, Invitae Hypoglycemia Panel). The coding regions and exon–intron boundaries of the tested genes, including *ABCC8*, were analyzed. All suspected variants were confirmed using Sanger sequencing. In order to identify the origin of the mutations, sequencing of the relevant exons was also performed in DNA samples from the patients’ parents. CNVs, defined as single exon or larger deletions or duplications (Dels/Dups), were detected from the sequence analysis data using a proprietary bioinformatics pipeline. If a CNV was identified, the multiplex ligation-dependent probe amplification (MLPA) technique (MRC-Holland, Amsterdam, Holland) or MLPA-seq was run to confirm the variant. The manual classification of detected mutations was performed in accordance with the American College of Medical Genetics and Genomics (ACMG) variant classification guidelines as well as the gnomAD, ClinVar, HGMD Professional and Alamut Visual databases. In addition, the clinical relevance of any identified CNVs was evaluated by reviewing the relevant literature and databases, such as the Database of Genomic Variants and DECIPHER.

The obtained results were comparable with those present in the specialized databases (ClinVar and gnomAD), appeared to be consistent with the data presented in similar studies and revealed the importance of genetic testing in achieving early diagnosis and management and prognosis in the case of CHI patients.

This study was conducted according to the Declaration of Helsinki, and it was approved by the Ethics Committee of the Children’s Emergency Clinical Hospital, St. Maria Iași, Romania (Certificate no 7485/5 March 2024). In the case of both patients, the informed consent of the parents was obtained as well as of the adults who were clinically evaluated and genetically tested.

## 6. Conclusions

CHI is a major cause of hypoglycemia in the neonatal and childhood periods. Early diagnosis and appropriate management of HH are important to avoid long-term neurological complications.

The use of (18)F-DOPA PET/CT to differentially diagnose diffuse from focal CHI has completely changed the approach to diagnosis and management in these patients in recent years. For the future, the management of diffuse CHI that does not respond to drug treatment remains a challenge, and the identification of genetic mechanisms will provide new insights into the physiology of pancreatic β cells. Also, genetic testing must be included in the management of patients with HH, as there is a correlation between the genetic mutation and the clinical manifestations.

The two presented cases illustrate variable phenotypes (diffuse/focal) in patients with HH caused by different mutations in the *ABCC8* gene. They demonstrate the importance and clinical utility of genetic analysis for diagnosis and treatment guidance.

In both cases, the hypoglycemia started in the first days of life, and treatment with diaxoxide was initiated. Both patients were unresponsive to this treatment, requiring the change of diazoxide to somatostatin. Patients with heterozygous mutations in *ABCC8* and their family members require long-term monitoring, as they are at increased risk of developing diabetes mellitus.

## Figures and Tables

**Figure 1 ijms-25-05533-f001:**
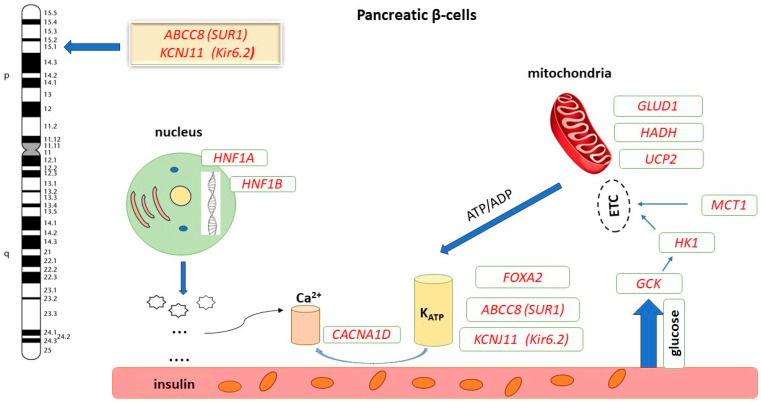
Genetic heterogeneity and molecular pathways in hyperinsulinism [5]. *ABCC8*: ATP-binding cassette, subfamily C, member 8; *KCNJ11*: Potassium channel, inwardly rectifying, subfamily J, member 11; *GCK*: Glucokinase gene; *HK1*: Hexokinase 1 gene; *SLC16A1/MCT1*: Solute carrier family 16 (monocarboxylic acid transporter), member 1 gene; *GLUD1*: Glutamate dehydrogenase 1 gene; *HADH*: 3-hydroxyacyl-CoA dehydrogenase gene; *UCP2*: Uncoupling protein gene; *CACNA1D*: Calcium channel, voltage-dependent, L type, alpha-1d subunit gene; *FOXA2*: Forkhead box A2; *HNF1A:* HNF1 homeobox A; ETC: Electron transport chain.

**Figure 2 ijms-25-05533-f002:**
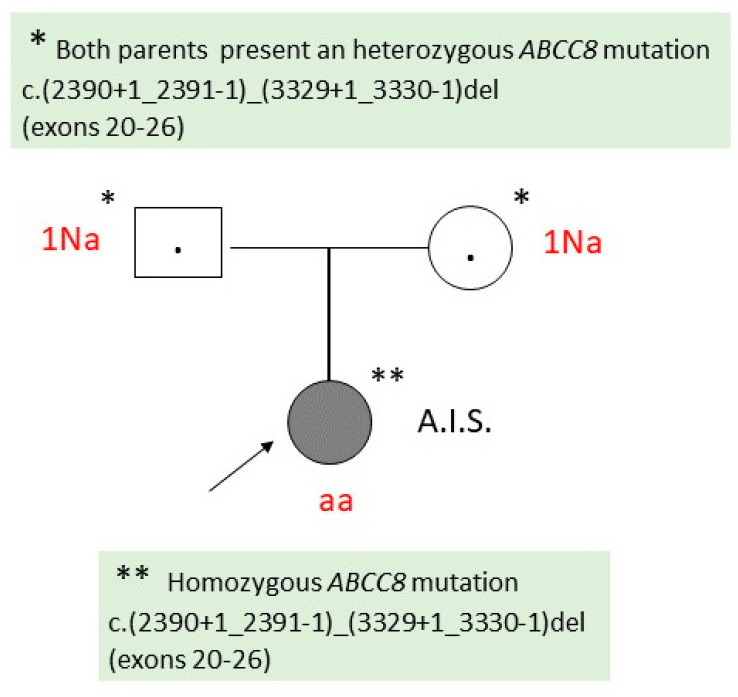
Family tree of patient 1 (A.I.S.). Na: Heterozygous genotype (healthy carrier); aa: Homozygous genotype (affected individual).

**Figure 3 ijms-25-05533-f003:**
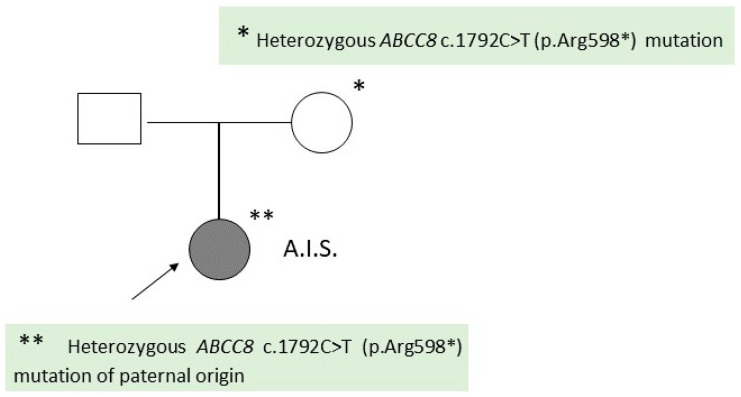
Family tree of patient 2 (D.M.S.).

**Table 1 ijms-25-05533-t001:** Clinical and paraclinical data of patients with congenital hyperinsulinemic hypoglycemia.

Criteria	Patient A.I.S.	Patient D.M.S.
Histologic type	Diffuse CHI	Focal CHI
The result of the patient’s genetic testing	Homozygous *ABCC8* c.(2390+1_2391-1)_(3329+1_3330-1)del (exons 20-26)	Heterozygous *ABCC8* c.1792C>T (p.Arg598*)
The result of genetic testing of the patient’s parents	Both parents: Heterozygous *ABCC8* c.(2390+1_2391-1)_(3329+1_3330-1)del (exons 20–26)	Father: Heterozygous *ABCC8* mutation c.1792C>T (p.Arg598*) Mother: normal result
Gender	F	F
Family history	no	no
Gestation	Term (38 weeks)	Term (38 weeks)
Parents’ consanguinity	No	No
Birth weight	3140 g	2700 g
Onset of symptoms	1st day	3rd day
Persistent	13–32 mg/dL	17–45 mg/dL
Insulin plasma level	↑ (41.45 uIU/mL; normal value: 3–25 uIU/mL)	15.28 uIU/mL (normal value: 3–25 uIU/mL)
C-peptide plasma level	↑ (4.92 ng/mL; normal value: 0.2–4.4 ng/mL)	2.2 ng/mL (normal value: 0.9–7.1 ng/mL)
hGH	↑ (133.67 uIU/mL; normal value: 0–5 uIU/mL)	Normal value
Thyroid hormones	Normal value	Normal value
Cortisol plasma level	Normal value	↓ (2.27 µg/dL; normal value: 4.3–22.4 µg/dL)
ACTH	Not performed	ACTH < 5 pg/mL (normal value: 5–46 pg/mL)
Macrosomia	No	No
Neurological manifestations	No	Tonic-clonic seizures Generalized hypotonia
Perinatal asphyxia	Yes	No
Transfontanelle ultrasonography	Bilateral subependymal hemorrhage	Not performed
Brain CT	Not performed	Pathologic
Abdominal MRI	normal	Normal
EEG	normal	Hypsarrhythmia
Diazoxide responsiveness	No	No
(18)F-DOPA PET/CT	Not indicated	Not performed yet

C-peptide, insulin, cortisol collected at hypoglycemia values below 50 mg/dL; ACTH: Adrenocorticotropic hormone; CT: Computed tomography; EEG: Electroencephalogram; (18)F-DOPA PET/CT: (3,4-dihydroxy-6-[^18^F]fluoro-L-phenylalanine) positron emission tomography scan/computed tomography; hGH: Human growth hormone; MRI: Magnetic resonance imaging; PET; Positron emission tomography scan.

## Data Availability

Data are contained within the article.

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
