# Peer review of "Congenital Hyperinsulinism Caused by Mutations in ABCC8 Gene Associated with Early-Onset Neonatal Hypoglycemia: Genetic Heterogeneity Correlated with Phenotypic Variability"

_ijms, 2024, doi:10.3390/ijms25105533_

Round 1

Reviewer 1 Report

Comments and Suggestions for Authors

Your manuscript, " Genetic Heterogeneity Correlated with Phenotypic Variability in Congenital Hyperinsulinism Caused by Mutation in ABCC8 Gene Associated with Early-Onset Persistent Neonatal Hypoglycemia ", provides a solid foundation for understanding the significance of your research. Article may be accepted after major revision. I would like to comment:

1:  Title should be concise and revised

2. Author has find mutations in two cases. In first case homozygous mutation and in sec case he found heterozygous mutations. Both these mutations should be clearly describe

3. The methodology section is poorly described. No complete methodology provided

4. Author did not explain what is the link between treatment of diaxoxide/ somatostatin with mutation or with current study

Comments on the Quality of English Language

English is good

Author Response

Thank you very much for taking the time to review this manuscript. Please find the detailed responses below and the corresponding revisions/corrections highlighted (red color) in the re-submitted files.

  1. Title should be concise and revised

R: Thank you for your recommendation. I change the title as reformulated by the second reviewer:

"Congenital Hyperinsulinism Caused by Mutations in ABCC8 Gene Associated with Early-Onset Neonatal Hypoglycemia: Genetic Heterogeneity Correlated with Phenotypic Variability"

  1. Author has find mutations in two cases. In first case homozygous mutation and in sec case he found heterozygous mutations. Both these mutations should be clearly describe

R: Thank you for your recommendation. I have completed the description of the two mutations in the new version of the article (marked with red color in the text).

  1. The methodology section is poorly described. No complete methodology provided

R: Thank you for your recommendation. I have completed the description of the methods used at the two laboratories (Bluprint Genetics and Invitae) (marked in red color in the text).

  1. Author did not explain what is the link between treatment of diaxoxide/ somatostatin with mutation or with current study

R: Thank you for your recommendation. In the discussion chapter, I completed the aspects related to the relationships between the type of mutations identified and the response to treatment with diazoxide, making comparisons with data from the literature. (marked in red color in the text).

Please check the new version of the article that I have attached. Thank you once again for the effort you put into revising our article.

Dr. Butnariu

Reviewer 2 Report

Comments and Suggestions for Authors

To correlate genetic heterogeneity with phenotypic variability, the authors presented 2 patients with early-onset hyperinsulinemic hypoglycemia (HH), caused by different ABCC8 gene mutations, indicating that molecular genetic testing could play an important role for diagnosis, management and genetic counseling for HH patients.

(I) Major Comments

This manuscript tried to pinpoint genetic mutations for congenital hyperinsulinism, also referred to as hyperinsulinemic hypoglycemia (HH), a genetically heterogeneous disease that constitutes the most common form of severe and persistent hypoglycemia during infancy and childhood, and  generally speaking, the atuhors have presented new information on genetic causes of HH.

I have the following major comments:

(1) Page 1, Manuscript title,

"Genetic Heterogeneity Correlated with Phenotypic Variability in Congenital Hyperinsulinism Caused by Mutation in ABCC8 Gene Associated with Early-Onset Persistent Neonatal Hypoglycemia"

could be corrected to

"Congenital Hyperinsulinism Caused by Mutations in ABCC8 Gene Associated with Early-Onset Neonatal Hypoglycemia: Genetic Heterogeneity Correlated with Phenotypic Variability"

In the above corrected, "ABCC8" could be in italic font. The revised title makes the title clearer and easier for the audience to read.

(2) Page 1, Abstract, lines 28-29,

"with early-onset hyperinsulinemic hy-poglycemia caused by"

could be corrected to

"with early-onset HH caused by"

The reason for the above corrected is that the term "hyperinsulinemic hypoglycemia" has been abbreviated as "HH" earlier in this paragraph on Page 1, line 25.

(3) Page 3, lines 123-125,

"(normal values 3-25 u IU/mL) and C peptide of 4.92 ng/mL (normal values 0.2-4.4 ng/ mL) - both increased, serum cortisol and thyroid hormones were within normal limits with  growth hormone increased 133.67 uUI/ml (normal values 0-55 uUI/mlL) (table 1)."

could be corrected to

"(normal value: 3-25 uIU/mL) and C peptide of 4.92 ng/mL (normal value: 0.2-4.4 ng/ mL) - both increased, serum cortisol and thyroid hormones were within normal limits with an increased growth hormone of 133.67 uUI/ml (normal value: 0-55 uUI/mlL) (Table 1)."

(4) Page 4 line 126, Table 1's title,

"Congenital Hyperinsulinemic hypoglycemia."

could be corrected to

"Congenital Hyperinsulinemic Hypoglycemia."

As shown in above corrected, "hypoglycemia" has been corrected to "Hypoglycemia"

(5) Page 4, Table 1, 1st column (i.e., column header is "Criteria"), line 10,

"Persistent hypoglycaemia"

could be corrected to

"Persistent hypoglycemia"

(6) Page 4, Table 1, 2nd column (i.e., column header is "Patient A.I.S."), line 10,

"13-32mg/dL"

could be corrected to

"13-32 mg/dL"

As shown in above corrected, there shall be a blank space between "13-32" and "mg/dL".

(7) Page 4, Table 1, 2nd column (i.e., column header is "Patient A.I.S."), line 11,

"(41,45 uUI/ mL; normal value: 3-25 uUI/ mL)"

could be corrected to

"(41.45 uUI/mL; normal value: 3-25 uUI/mL)"

As shown in above corrected, "41,45 uUI/ mL" has been corrected to "41.45 uUI/mL", and all occurrences of "/ mL" shall be corrected to "/mL".

(8) Page 4, Table 1, 2nd column (i.e., column header is "Patient A.I.S."), line 12,

"(4,92 ng/ mL; normal value : 0,2-4.4 ng/ mL)"

could be corrected to

"(4.92 ng/mL; normal value: 0.2-4.4 ng/mL)"

As shown in above corrected, "ng/ mL" has been corrected to "ng/mL", "normal value :" bas been corrected to "normal value:", and all usages of "," to denote decimal point shall be corrected to the usages of "." to denote decimal point (i.e., 

"4,92" is corrected to "4.92", and "0,2" is corrected to "0.2").

(9) Page 4, Table 1, 2nd column (i.e., column header is "Patient A.I.S."), line 13,

"(133,67 uUI/mL ; normal value: 0-5 uUI/ mL)"

could be corrected to

"(133.67 uUI/mL; normal value: 0-5 uUI/mL)"

As shown in above corrected, "uUI/ mL" has been corrected to "uUI/mL", and all usages of "," to denote decimal point shall be corrected to the usages of "." to denote decimal point (i.e., 

"133,67" is corrected to "133.67").

(10) Page 4, Table 1, 3rd column (i.e., column header is "Patient D.M.S."), line 8,

"2700g"

could be corrected to

"2700 g"

As shown in above corrected, there shall be a blank space between "2700" and "g".

(11) Page 4, Table 1, 3rd column (i.e., column header is "Patient D.M.S."), line 11,

"15.28 uUI/ ml (normal value : 3-25 uUI/ml )"

could be corrected to

"15.28 uUI/ml (normal value: 3-25 uUI/ml)"

As shown in above corrected, "uUI/ mL" has been corrected to "uUI/mL", and all occurrences of "normal value :" shall be corrected to "normal value:".

(12) Page 4, Table 1, 3rd column (i.e., column header is "Patient D.M.S."), line 12,

"2,2 ng/ ml (normal value: 0,9-7,1 ng/mL"

could be corrected to

"2.2 ng/ml (normal value: 0.9-7.1 ng/mL)"

As shown in above corrected, all usages of "," to denote decimal point shall be corrected to the usages of "." to denote decimal point, and a right parenthesis ")" has been added.

(13) Page 4, Table 1, 3rd column (i.e., column header is "Patient D.M.S."), line 15,

"( 2,27 μg/ dL( normal value: 4,3-22,4 μg/dL )"

could be corrected to

"(2.27 μg/dL; normal value: 4.3-22.4 μg/dL)"

As shown in above corrected, "μg/ dL(" has been corrected to " μg/dL;", and all usages of "," to denote decimal point shall be corrected to the usages of "." to denote decimal point, and a right parenthesis ")" has been added.

(14) Page 4, Table 1, 3rd column (i.e., column header is "Patient D.M.S."), line 16,

"ACTH < 5 pg/mL (normal value: 5 -46 pg/ mL)"

could be corrected to

"ACTH < 5 pg/mL (normal value: 5-46 pg/mL)"

As shown in above corrected, "5 -46(" has been corrected to "5-46", and all occurrences of "pg/ mL" shall be corrected to "pg/mL".

(15) In Table 1's footnote, the authors shall state "Abbreviations: ", and then, add the full terms for the following abbreviated terms (in alphabetical order): ACTH, CT, (18)F-DOPA, EEG, hGH, IRM, and PET.

(16) Page 5, in Figure 2, 

In the "**" text, "ABCC8" should be in italic font.

(17) Page 5, in Figure 3, 

In the "**" text, "ABCC8" should be in italic font.

(18) Page 5, line 147,

"Abdominal MRI (Magnetic Resonance Imaging)"

could be corrected to

"Abdominal Magnetic Resonance Imaging (MRI)"

(19) Page 6, line 199,

"At the age of 1 month and 10 days"

could be corrected to

"At the ages of 10 days and 1 month,"

As shown in above, "age" has been corrected to "ages", because there are 2 different ages indicated, and "10 days" shall be mentioned first because this occurred before "1 month".

(20) Page 7, line 244,

"The two cases presented reflect genetic heterogeneity"

could be corrected to

"The two cases presented demonstrated genetic heterogeneity"

(21) Page 7, lines 252-253,

"according to The Genome Aggregation Database (gnomAD; dbSNP rs139328569) (http://gnomad.broadinstitute.org, accessed on 7 March 2024) [18]"

could be corrected to

"according to The Genome Aggregation Database (gnomAD; http://gnomad.broadinstitute.org, accessed on 7 March 2024) [18], for this SNP with dbSNP ID rs139328569."

(22) Page 8, lines 302-304,

"noninsulin-dependent (OMIM 125853), permanent neonatal (PNDM) (OMIM 606176) and transient neonatal type 2 diabetes mellitus (OMIM 610374)."

could be corrected to

"type 2 diabetes mellitus (OMIM 125853), PNDM (OMIM 606176) and transient neonatal diabetes mellitus (OMIM 610374)."

(23) Page 8, line 348,

"4. Genetic counseling"

could be corrected to

"4. Genetic Counseling"

(24) Page 8, line 361,

"5. Material and method"

could be corrected to

"5. Materials and Methods"

(25) Page 9, lines 367-370,

"The obtained re-sults were compared with those present in the specialized literature ( Clinvar and The Genome Aggregation Database - gnomAD ) being consistent with the data presented in similar studies,"

could be corrected to

"The obtained results were comparable with those present in the specialized databases (ClinVar and gnomAD), and appeared to be consistent with the data presented in similar studies,"

(II) Minor Comments

First of all, in the main text consisting of "1. Introduction", "2. Results", "3. Discussion", "4. Genetic Counseling", "5. Materials and Methods", and "6. Conclusions",

for an abbreviated term, in that term’s first appearance, the authors shall spell out the full term and then put the abbreviated term in a pair of parentheses, and afterwards, only the abbreviated term could be used, rather than the full term, e.g.,

Page 1, lines 36-37,

"PET/CT was recommended (an investigation that cannot be carried out in the country)"

could be corrected to

"The positron emission tomography/computed tomography (PET/CT) was recommended (an investigation that cannot be carried out in the country)"

In addition, the following grammatical and typographical errors should be corrected:

(1) Page 1, line 23,

"[email protected]; (D.A.B.)"

could be corrected to

"[email protected] (D.A.B.)"

(2) Page 1, lines 43-44,

"ATP- sensitive potassium channel"

could be corrected to

"ATP-sensitive potassium channel"

(3) Page 2, lines 48-49,

"severe and persistent hypoglycaemia (Hyperinsulinaemic Hypoglycaemia, HH)"

could be corrected to

"severe and persistent hypoglycemia (Hyperinsulinemic Hypoglycemia, HH)"

As shown in above corrected, the authors shall use the standard term "hyperinsulinemic" and the standard term "hypoglycemia" consistently, rather than other spelling forms.

(4) Page 2, line 71,

"an (18)F-DOPA PET/CT"

could be corrected to

"an (18)F-dihydroxyphenylalanine (18F-DOPA) positron emission tomography/computed tomography (PET/CT) "

(5) Page 4, lines 128-129,

"positron emission tomography scan/ com-puted tomography"

could be corrected to

"positron emission tomography scan/computed tomography"

As shown in above, there shall not blank space after "scan/".

(6) Page 6, line 174,

"because the child refusal to eat,"

could be corrected to

"because the child refused to eat,"

(7) Page 6, lines 179-181,

"(normal values: 3-25 uIU/mL) and C-peptide 2.2 ng/mL (normal values: 0.9-7.1 ng/mL), respectively 2.01 ng/mL, in two consecutive days (table 1)."

could be corrected to

"(normal value: 3-25 uIU/mL) and C-peptide 2.2 ng/mL (normal value: 0.9-7.1 ng/mL), respectively, in two consecutive days (Table 1)."

As shown in above, all occurrences of "normal values:" shall be corrected to "normal values".

Also, please double check the above to ensure accuracy.

(8) Page 6, line 184,

"(normal values: 4.3-22.4 mcg/mL)"

could be corrected to

"(normal value: 4.3-22.4 mcg/mL)"

As shown in above, all occurrences of "normal values:" shall be corrected to "normal values".

(9) Page 6, line 185,

"(normal values: 5 - 46 pg/mL)"

could be corrected to

"(normal value: 5-46 pg/mL)"

As shown in above, all occurrences of "normal values:" shall be corrected to "normal values", and "5 - 46" has been corrected to "5-46", respectively.

(10) Page 7, line 241,

"-2.16 SD"

could be corrected to

"-2.16 standard deviation (SD)"

(11) Page 7, line 261,

"[21,23].This variant"

could be corrected to

"[21,23]. This variant"

As shown in above corrected, there should be a blank space between "." and "this".

(12) Page 7, line 266,

"type 1) -associated variant inherited"

could be corrected to

"type 1)-associated variant inherited"

(13) Page 8, line 296,

"are associated with hyperinsulinemic hypoglycemia"

could be corrected to

"are associated with HH"

(14) Page 8, line 300,

"familial hyperinsulinemic hypoglycemia, type 1 (FHI) (OMIM 256450)"

could be corrected to

"FHI (OMIM 256450)"

The reason for the above corrected is that the term "familial hyperinsulinemic hypoglycemia, type 1" has been abbreviated as "FHI" earlier in the main text on Page 7, line 265.

(15) Page 8, line 308,

"Gain-of function (GOF)"

could be corrected to

"Gain-of-function (GOF)"

(16) Page 10, line 385,

"with hyperinsulinemic hypoglycemia caused by different mutations in the . gene."

could be corrected to

"with HH caused by different mutations in the gene."

(17) Page 10, line 422,

"PNDM: Permanent neonatal type 2 diabetes mellitus;"

could be corrected to

"PNDM: Permanent neonatal diabetes mellitus;"

(18) Page 10, line 423,

"LOF: Loss-of function; GOF: Gain-of function"

could be corrected to

"LOF: Loss-of-function; GOF: Gain-of-function"

(19) Page 10, line 412,

"HH : Hyperinsulinemic hypoglycemia"

could be corrected to

"HH: Hyperinsulinemic hypoglycemia"

The above are just several examples, and the authors shall perform a careful and thorough checking on the main text to correct all errors.

Comments on the Quality of English Language

Moderate editing of English language required

Author Response

Thank you very much for taking the time to review this manuscript. Please find the detailed responses below and the corresponding revisions/corrections highlighted (red color) in the re-submitted files.

  • Page 1, Manuscript title:

R: Thank you for your recommendation. I changed the title as you reformulated it.

"Congenital Hyperinsulinism Caused by Mutations in ABCC8 Gene Associated with Early-Onset Neonatal Hypoglycemia: Genetic Heterogeneity Correlated with Phenotypic Variability"

(2.) – (19)  

R: Thank you for your recommendation. I have corrected all the errors you reported (all  changes are marked in the text with red color).

Please check the new version of the article that I have attached. Thank you once again for the effort you put into revising our article.

Dr. Butnariu

Round 2

Reviewer 1 Report

Comments and Suggestions for Authors

Thank you for sharing modified version of article. The author has modified it as per suggestions. Article may be accepted in current form.

Author Response

Thank you very much for taking the time and for the effort to review this manuscript. Please check the new version of the article that I have attached. 

Reviewer 2 Report

Comments and Suggestions for Authors

In this revised manuscript, Revision 1, the authors have addressed a majority of reviewer comments, but there are the following further comments that the authors shall address adequately, and when the authors submit a newly revised manuscript, please accept all previous changes, and therefore, do not have to track changes and can put the text color to normal black ink color, and only for changes for addressing the following major and minor comments, please put the text color in red font color (i.e., the red font color will indicate the changes made):

(I) Major Comments

(1) Page 8, lines 276-280,

"This variant (Arg598Ter) was also reported as pathogenic in ClinVar (Variation ID: 434056) [22]. Of the 11 affected individuals, at least 4 were compound heterozygotes car-rying a reported pathogenic variant in trans, increasing the likelihood that the Arg598Ter variant is pathogenic (Variation ID: 434053) [23-27]"

could be corrected to

"This variant (Arg598Ter) was also reported as pathogenic in ClinVar (Variation ID: 434056) (https://www.ncbi.nlm.nih.gov/clinvar/variation/434056/, accessed on 7 March 2024) [22]. Of the 11 affected individuals, at least 4 were compound heterozygotes carrying a reported pathogenic variant in trans, increasing the likelihood that the Arg598Ter variant is pathogenic [23-27]"

As shown in above corrected, the "Variation ID: 434053" (i.e., URL: https://www.ncbi.nlm.nih.gov/clinvar/variation/434053/ ) refers to "p.Arg998Ter" (which is not "p.Arg598Ter"), which is incorrect, and should be deleted

(2) Page 10, lines 413-414,

"he will also have somatic paternal UPD for chromosome 11p15.5 is low [19]."

could be corrected to

"he will also have somatic paternal uniparental disomy (UPD) for chromosome 11p15.5 is low [Flanagan SE, et al., Paternal Uniparental Isodisomy of Chromosome 11p15.5 within the Pancreas Causes Isolated Hyperinsulinemic Hypoglycemia. Front Endocrinol (Lausanne). 2011;2:66. PMID: 22654821]."

As shown in above corrected, the Reference 19, i.e., Banerjee I, et al., Therapies and outcomes of congenital hyperinsulinism-induced hypoglycaemia. Diabet Med. 2019;36:9-21. PMID: 30246418, is incorrect, and the correct reference, i.e., Flanagan SE, et al., Front Endocrinol (Lausanne). 2011;2:66. PMID: 22654821 has been added, for which the authors shall correct the References section accordingly

Also, "UPD" has been corrected to "uniparental disomy (UPD)"

(II) Minor Comments

First of all, in the main text consisting of "1. Introduction", "2. Results", "3. Discussion", "4. Genetic Counseling", "5. Materials and Methods", and "6. Conclusions",

all occurrences of "KIR6.2" shall be corrected to "Kir6.2", to use the standard term "Kir6.2" consistently in the main text.

The following grammatical and typographical errors should be corrected:

(1) Page 2, line 52,

"(KIR6.2) genes, located on"

could be corrected to

"(Kir6.2) genes, located on"

(2) Page 3, line 76,

"encode the SUR/KIR6"

could be corrected to

"encode the SUR1/Kir6"

(3) Page 3, line 107,

"We present the patient A.I.S,"

could be corrected to

"We present the patient A.I.S.,"

As shown in above corrected, the "A.I.S," has been corrected to "A.I.S.,"

(4) Page 4, line 144,

"somatostatin analogues (sandostatin)"

could be corrected to

"somatostatin analogue (sandostatin)"

As shown in above corrected, the "analogues" has been corrected to "analogue"

(5) Page 5, line 168,

"is also a girl, D.M.S,"

could be corrected to

"is also a girl, D.M.S.,"

As shown in above corrected, the "D.M.S," has been corrected to "D.M.S.,"

(6) Page 6, line 190,

"(normal values: 4.3-22.4 mcg/mL)"

could be corrected to

"(normal value: 4.3-22.4 mcg/mL)"

As shown in above corrected, the "normal values" has been corrected to "normal value"

(7) Page 6, line 200,

"somatostatin analogue (Sandostatin)"

could be corrected to

"somatostatin analogue (sandostatin)"

As shown in above corrected, the "Sandostatin" has been corrected to "sandostatin"

(8) Page 7, line 256,

"del/dup (CNV) analysis using"

could be corrected to

"del/dup copy number variation (CNV) analysis using"

(9) Page 8, line 274,

"for this SNP with dbSNP ID rs13932856"

could be corrected to

"for this single nucleotide polymorphism (SNP) with National Center for Biotechnology Information (NCBI) dbSNP ID rs13932856"

(10) Page 8, lines 286-287,

"a single heterozygous pathogenic FHI (familial hyperinsulinemic hypo-glycemia, type 1)-associated variant"

could be corrected to

"a single heterozygous pathogenic familial hyperinsulinemic hypoglycemia, type 1 (FHI)-associated variant"

(11) Page 9, line 333,

"intracellular calcium [Ca2+]i"

In above, "[Ca2+]i", the "2+" shall be in superscript, and the "i" shall be in subscript, respectively.

(12) Page 9, line 353,

"cases with FHI-KATP have"

could be corrected to

"cases with FHI have"

(13) Page 10, line 400,

"The phenotype of heterozygous patients was"

could be corrected to

"The phenotypes of heterozygous patients were"

(14) Page 10, line 411,

"is 50%, correlated with autosomal dominant pattern"

could be corrected to

"is 50%, which correlates with autosomal dominant pattern"

(15) Page 11, lines 419-420,

"two patients with hyperinsuline-mic hypoglycemia diagnosed"

could be corrected to

"two patients with HH diagnosed"

(16) Page 11, lines 433-434,

"Copy number variations (CNVs), defined as single exon or larger deletions or duplications (Del/Dups),"

could be corrected to

"CNVs, defined as single exon or larger deletions or duplications (Dels/Dups),"

As shown in above corrected, the "Copy number variations (CNVs)" has been corrected to "CNVs", and "(Del/Dups)" has been corrected to "(Dels/Dups)", respectively

(17) Page 11, line 447,

"according to the Helsinki II declaration"

could be corrected to

"according to the Declaration of Helsinki"

(18) Page 12, line 501,

"DCM: Disease-causing mutation (DCM);"

could be corrected to

"DCM: Disease-causing mutation;"

Comments on the Quality of English Language

Moderate editing of English language required

Author Response

Thank you very much for taking the time and for the effort to review this manuscript. I made all the changes you suggested  (marked in red in the text). Please check the new version of the article that I have attached.